# Impact of the grassland ecological compensation policy on pastoral production efficiency— evidence from pastoral China

**Fang Ju**©, **Wenjie Ouyang**©, **Chengtao Zhang, Jianjun Zhang**◉*

College of Economics and Management, Inner Mongolia Agricultural University, Hohhot, China

© These authors contributed equally to this work.
* jgyzjj@imau.edu.cn

## Abstract

The impact of the Grassland Ecological Compensation Policy (GECP), which protects grassland ecology while promoting the transformation and improvement of grassland animal husbandry, on pastoral production efficiency is among the important indicators for evaluating the sustainable development of animal husbandry, and it directly affects the economic benefits of herders. In this study, based on field questionnaire data from 468 herders in the pastoral areas of the Inner Mongolia Autonomous Region of China, a two-stage semi-parametric DEA model and a mediated effect model were used to measure herders' pastoral production efficiency and analyze the effect and influence mechanism of the GECP on herders' pastoral production efficiency, respectively. The results of the study show that (1) the overall pastoral production efficiency of herders in Inner Mongolia is low, with a mean value of 0.43, which is caused mainly by low pure technical efficiency. (2) The factors that have a significant positive effect on the herding efficiency include the number of years of education of the head of the household, whether the head of the household has received technical training, the percentage of herding income from herding, the number of herding machines available, the evaluation of herding socialized services and whether pasture is rented or owned. (3) The realization of appropriate-scale management by herders plays a role in mediating herders' pastoral production efficiency. The realization of moderate-scale management by herders has a partial intermediary effect on the process through which the GECP affects herders' pastoral production efficiency, and the direct effect of this policy on the pastoral production efficiency of large-scale herders is significantly greater than that on small-scale herders. Based on the above conclusions, suggestions are proposed for improving support policies that offer ecological bonuses, demonstrating and popularizing breeding technology, and guiding moderate-scale management.

**Data availability statement:** The research data and code used in this article can be found in Supplementary Information.

**Funding:** This research was funded by the National Natural Science Foundation of China Project "Study on the Influence Mechanism of Grassland Ecological Compensation and Reward Policy on Part-time behavior of Grazing Herdsmen——Taking Inner Mongolia as an Example" (72363025) The roles played by funders in research include design, data collection and analysis, and decision-making for publication.

**Competing interests:** The authors have declared that no competing interests exist.

## 1. Introduction

Globally, the overexploitation of ecosystems remains a serious problem that needs to be resolved through human development, and how to harmonize the promotion of ecological environmental protection and the enhancement of people's well-being has become the focus of research conducted by national governments [1]. In this context, payment for ecosystem services (PES) programs have received widespread attention as an incentive-based environmental policy tool in areas where ecologically fragile zones overlap with poverty zones [2,3]. The core logic is that, based on the principle of "who protects, who benefits", the external effects of ecological services are internalized through the economic compensation mechanism to achieve the dual goals of ecological protection and livelihood improvement in a synergistic manner [4]. Over the past few decades, this model has evolved into a policy consensus among various countries. In Switzerland, the Ecological Compensation Area Policy (ECA) encourages farmers to set aside ecological compensation areas on their farms by granting them subsidies, which in turn promotes biodiversity restoration [5]. In the United States, the Conservation Reserve Program (CRP) leaves environmentally sensitive agricultural land fallow, which reduces soil erosion, improves water quality, increases carbon sinks, and protects ecological and environmental security [6].

In China, grassland is among the most important terrestrial ecosystems and is an important vehicle for livestock production and a means of production on which herders depend. However, under the combined influence of extreme climatic and anthropogenic factors, grasslands have been degraded to varying degrees, seriously threatening ecological security and the livelihoods of herders [7]. To this end, in 2011, China began to implement a large-scale payment for ecosystem services program, the Grassland Ecological Conservation Subsidy and Incentive Mechanism Policy (hereinafter referred to as the "Grassland Ecological Compensation Policy (GECP)), in the major grassland pastoral provinces (regions) on a five-year cycle that has now reached its third round. The aim of this policy is to guide herders to reduce livestock to protect the ecology and realize the sustainable development of grassland and animal husbandry by providing herders with subsidies for grazing bans, incentives for grass–animal balance and subsidies for pastoral production materials [8]. The framework of forbidden grazing and grass–animal balance is the core of this policy; "forbidden grazing" refers to the implementation of year-round forbidden grazing and closed cultivation for severely degraded pastures, and "grass–animal balance" refers to the government's rationalization of the livestock carrying capacity based on the size of the pasture and the ecological situation [9]. After the implementation of the policy, the farming methods of herders have gradually changed from free grazing to "barn feeding" or "semi confinement", and there has been a marked shift in the production methods of the pastoral industry, with a concomitant change in the pastoral production efficiency of herders. Therefore, revealing the mechanism by which the GECP affects the pastoral production efficiency of herders can not only provide empirical support for addressing the dual problems of ecological protection in grassland pastoral areas and improvement of herders' livelihoods but also effectively

motivate herders to improve their pastoral production efficiency and continue to promote the transformation and improvement of grassland animal husbandry.

## 2. Literature review

A review of the literature on the implementation effects of payment for ecosystem services (PES) programs reveals two primary findings. First, the ecological impact of PES programs is notable. For instance, Thang (2021) employed microsurvey data to evaluate the environmental effects of Vietnam's PES program and reported that its implementation significantly improved natural forest conditions in the study area [10]. Conversely, Humberto (2023) utilized macro data to assess the effectiveness of PES programs in protecting forest cover in tropical regions of Mexico, revealing that the current PES methods were not highly effective [11]. Second, the economic impact of PES programs on the production activities of farmers and herders is significant. Seema (2023) demonstrated through a study of the economic benefits of a forest protection payment project in Uganda that PES programs play a crucial role in increasing the economic well-being of local residents [12]. Additionally, Orlando (2022) examined the relationship between PES programs and coffee production in Colombia and reported that PES programs have significantly improved the production efficiency of coffee producers [13]. Similarly, discussions on the implementation effects of the GECP in existing studies focus mainly on two aspects. On the one hand, assessment of the ecological effect of the policy, focusing on the systematic change in the key indicator of the natural grassland–normalized difference vegetation index (NDVI) [14–16], revealed that after the implementation of the GECP, the comprehensive vegetation cover, fresh grass yield and unit biomass of natural grassland improved, and the ecological carrying capacity significantly increased [17]. However, these ecological effects showed significant spatial heterogeneity; i.e., the policy had a greater impact on the quality of grassland in wet areas than in dry areas [18]. On the other hand, assessment of the economic effect of the policy, focusing mainly on changes in the income of herders [19], revealed that the GECP not only significantly increased the total income of low-income herders through the direct payment of ecological compensation [20] but also indirectly increased the total income of herders through the transfer of labor and expansion of herd size. In addition, the study found that regions with high levels of per capita income among herders benefited more from the policy, which indicates that the policy widened the development gap within the region [21]. Pastoral production efficiency, which is an important indicator for evaluating the sustainable development of animal husbandry, not only characterizes the utilization of natural pasture in the process of pasture production but also directly reflects the economic benefits to herders. By studying the impact of the GECP on the pastoral production efficiency of herders, it is possible to reveal the impact of the policy on pastoral production. Some studies have reported that after the implementation of the GECP, the growth of productivity of professional sheep fattening farming and professional beef fattening farming in pastoral areas has been stable [22,23]. However, other studies have reported that ecological compensation has difficulty compensating for the increase in the cost of shepherding as a result of the policy, and the greater the duration of the ban on animal husbandry is, the greater the decrease in the efficiency of herders' technological progress [24]. A significant reason for these disputes is the varying methodologies employed by different studies to measure pastoral production efficiency. Currently, the majority of studies rely on relatively straightforward efficiency evaluation methods, such as the single-stage DEA model or Malmquist index analysis. However, due to the complexities inherent in policy implementation effects and the production behaviors of herders, these methods may have limitations in accurately reflecting changes in breeding efficiency, which in turn impacts the reliability of the conclusions drawn.

In summary, although existing research provides important theoretical support for this paper, several deficiencies still exist: (1) The existing research focuses mainly on the impact of the GECP on the ecological environment and the lives of herders, and few studies have focused on the impact of the GECP on the pastoral production efficiency of herders. (2) There is still considerable controversy over the current conclusions concerning the impact of the GECP on the pastoral production efficiency of herders. (3) Few studies have analyzed the mechanism by which the GECP affects the pastoral production efficiency of herders from the perspective of grassland-scale heterogeneity. In view of this, the marginal

contributions of this paper are reflected in the following three points: (1) By combining externality theory with technology induction theory, a theoretical analysis framework for the impact of the GECP on the pastoral production efficiency of herders is constructed, and the role of moderate-scale management in the relationship between the two is explored. (2) Based on field investigation data of herders in the pastoral areas of Inner Mongolia, China, the Two-Stage Double Bootstrap DEA model was used to calculate the pastoral production efficiency of herders, and the impact effect and impact path of the GECP on the pastoral production efficiency of herders were empirically tested. (3) The heterogeneity of the impact of the GECP on the breeding efficiency of herders with different grassland scales was explored. The aims of this study were to provide new ideas for increasing the pastoral production efficiency of herders, reducing the pressure on grassland ecological protection, improving the stability of herders' incomes and achieving high-quality development of animal husbandry.

This paper is organized as follows: Section 1 presents the Introduction. Section 2 presents the Literature Review. Section 3 presents the Materials and Methods, including the theoretical framework, data sources and model construction. Section 4 presents the Results. Section 5 provides the Discussion. Section 6 presents the Conclusions. Section 7 lists Recommendations.

## 3. Materials and methods

### 3.1. Theoretical analysis framework and research hypothesis

The GECP, by providing government supervision and granting subsidies, encourages herders to protect grassland ecology and drives the transformation and improvement of grassland animal husbandry [25]. First, due to the restriction of grazing, herders gradually changed their grazing methods from rough free grazing to "barn feeding" or "grazing + barn feeding", breaking the traditional grazing practices regarding pasture selection and livestock management, which are overly dependent on the natural environment and climatic conditions. According to the theory of "induced technological innovation", barn feeding encourages herders to adaptively learn new breeding techniques and management methods and feed their animals a scientifically based diet, which ultimately increases the rate of slaughter and turnover of livestock. Second, the policy has promoted the application and popularization of pasture machinery; in particular, the increase in the number of mixers and forage harvesting equipment, etc., is extremely obvious, and the increase in the level of infrastructure inputs has reduced labor inputs, providing better hardware conditions for the development of animal husbandry [26]. Finally, the use of pasture has gradually transformed to a combination of natural pasture and artificial forage land, the integration of grass and livestock has obviously strengthened, the system of rest and rotational grazing has improved the productivity of natural pasture, and the cultivation of high-quality pasture grasses such as shepherd's croft, oat and alfalfa has improved the self-sufficiency rate of fodder and ensured cold-season supplemental feeding to meet the nutritional needs of livestock while simultaneously reducing the cost of fodder [27]. In conclusion, the policy has positively impacted all aspects of herder production and improved pastoral production efficiency. In view of this, this paper proposes Hypothesis H1:

H1: The GECP significantly improves the pastoral production efficiency of herders.

The GECP promotes a reduction in a herder's herd size and the realization of moderate-scale management, which indirectly affect the herder's pastoral production efficiency. In the past, constrained by the scale of pastoral grassland animal husbandry and the degree of specialization of herders, the scale of a herder's breeding exceeded the carrying capacity of grassland ecology, resulting in the deterioration of grassland ecology, a slow increase in the income of herdsmen, and low economic efficiency of grassland animal husbandry [28]. One of the positive externalities of the implementation of the policy is that it prompts herders to implement moderate-scale management due to the policy's regulation of herders' livestock carrying capacity, narrowing the gap between the existing herd size and the optimal scale [29]. In the process of livestock reduction, the herder optimizes the allocation of breeding resources. On the one hand, the herder improves the quality of livestock, eliminating the "inferior" livestock in the population, introducing high-quality breeds of livestock for improvement, and carrying out "selective breeding" to improve the output capacity of the unit of livestock. On the other hand, the herder

optimizes the breeding structure, reduces the number of livestock with poor breeding efficiency according to the market situation, and even carries out single-species "fine breeding", ultimately resulting in the scale with the best operating efficiency to achieve moderate-scale management. Under a moderate scale of breeding, grassland ecosystems can reduce the adverse effects of climate change, fully release the production capacity of natural pastures, provide a sufficient forage supply for livestock, and reduce the pressure of competition for food among livestock to maintain good growth and reproductive capacity [30]. In view of this, Hypothesis H2 is proposed: (Fig 1)

H2: Realization of appropriate-scale management has a mediating effect on the impact path of the GECP on herders' pastoral production efficiency.

### 3.2. Household surveys and data collection

The data used in this study come from field surveys conducted by the research group from January to May 2024 in the cities of Hulunbeier and Chifeng, Xilingol League and Ordos in Inner Mongolia, China. The survey area was selected with the following considerations. First, the Inner Mongolia Autonomous Region is located in northern China and has a long and narrow shape, which is usually divided into three major parts, namely, the eastern, central and western parts, based on its physical geography, climate and economic characteristics. Therefore, to ensure that the sample reflects the actual situation of Inner Mongolia as much as possible, the research team chose Hulunbeier and Chifeng in the eastern region, Xilingol League in the central region, and Ordos in the western region, for a total of four leagues (cities) as the research area. Second, the research area of pastoral industry development is relatively aggregated and specialized. Because the main target of the GECP is herders, the selection of the region where pastoral production is the main industry as the research object can accurately reflect the impact of the GECP on the production efficiency of herders.

This study adopts a combination of stratified sampling, typical sampling and random sampling methods. First, stratified sampling methods are used to select 9 pastoralist banners (counties), such as New Balkhuru Rouge Banner, Arukolqin Banner and Wengniuot Banner, from the league (city) according to the annual per capita net income of the county. Second, typical sampling methods are used to select 1~3 soums (towns) and 2~3 gacha (villages) per soum (township). Finally, 12~17 farmers (villages) are selected in each gacha (village) by simple random sampling. In this study, 510 questionnaires for herders and 47 questionnaires for villages (gacha) were distributed in the grass–animal balance area, and 468 valid questionnaires for herders were obtained after the questionnaires with seriously missing data were excluded,

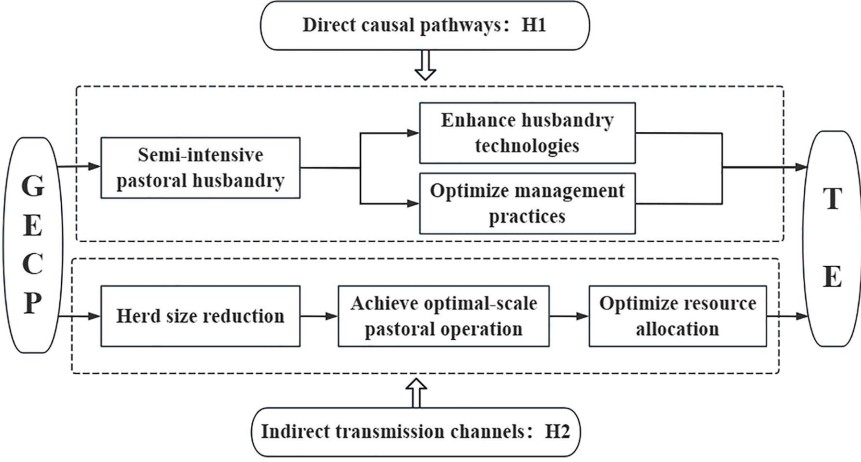

**Fig 1. Theoretical Framework of the Impact Mechanism of the GECP on Pastoral Production Efficiency.**

with an effective rate of 91.76%; 45 questionnaires for villages were valid, with an effective rate of 95.74%. The valid sample data cover 9 herding flags (counties), 19 soums (townships) and 45 villages (gacha).

## 3.3. Model methods

### 3.3.1. Two-stage double bootstrap DEA.

The Data Envelopment Analysis (DEA) model is a nonparametric measure for evaluating the relative effectiveness of decision-making units. The method determines the nonparametric segmented front surface of the data and measures the efficiency value by constructing different decision-making units using linear programming. There is no need to consider the form of the production function and the magnitude of the dimension, it is less affected by function setting errors, and it is possible to estimate efficiencies for multi-input data and multioutput data. The models include the Constant Returns to Scale (CRS) model and the Variable Returns to Scale (VRS) model, each of which can be subdivided into two categories: input-oriented and output-oriented [31].

Although most existing studies have used two-stage DEA (DEA+Tobit) to analyze the influencing factors of pastoral production efficiency, the method may invalidate statistical inferences about the magnitude of impacts on environmental variables based on efficiency estimates because of serial correlations of complex form and unknown structure between efficiency estimates. As a result, Simar and Wilson [32] proposed a two-stage semi-parametric DEA method (Two-Stage Double Bootstrap DEA), which is a hybrid method that combines nonparametric efficiency analysis (DEA) and statistical inference (bootstrap), and the core of the method is to correct the efficiency values through repeated sampling for the estimation bias and standard error of regression coefficients to more reliably analyze the influencing factors of efficiency. The approach is delineated as follows: (i) estimate the efficiency scores of each herder using Equation (2), (ii) estimate the truncated regression using Equation (4), (iii) calculate a set of bootstrap estimates, (iv) calculate each herder's TE score adjusted for bias, (v) use the TE scores adjusted for bias as the response variable in a truncated regression to estimate the factors affecting TE, (vi) conduct a series of bootstrapping to provide bootstrap estimates, and (vii) calculate new confidence intervals based on the bootstrap estimates. This method was chosen for this study to measure the pastoral production efficiency of herders in the first stage and to analyze the influencing factors in the second stage.

In the first stage of the DEA model selection process, Coelli et al. [33]argued that the CRS model is appropriate only if herders operate at an optimal scale and that incorrect assumptions about constant returns to scale may lead to inconsistent efficiency scores and consequently to a loss of statistical efficiency. In the pastoral areas of Inner Mongolia, competition with other herders and constraints regarding finances, pasture area and socioeconomic factors may prevent herders from realizing the optimal scale of their herds. Therefore, the hypothesis testing procedure for returns to scale proposed by Simar and Wilson [34] and Coelli [35] was used to test the herd size profile of herders in the study area. Specifically, the bootstrap statistic $\hat{\omega}$ was constructed to test the original hypothesis that "H0= overall herd technology scale reward is constant" and the alternative hypothesis that "H1= overall herd technology scale reward is variable". $\hat{\omega}$ is the statistic in Simar and Wilson (2002) Equation 4.6, which has been shown to be the most powerful of all statistics.

$$\hat{\omega} = \frac{\sum_{i=1}^{n} \theta_i^{CRS}}{\sum_{i=1}^{n} \theta_i^{VRS}}$$

(1)

In (Eq. 1), $\theta_i^{CRS}$ and $\theta_i^{VRS}$ are the pastoral production efficiency values of each herder calculated by the CRS model and VRS model, respectively, and the test $\hat{\omega}$ =0.854, P value = 0.03; therefore, the original hypothesis of "H0= overall herd technology scale reward is constant" is rejected, and the VRS model is selected in the first stage to measure the herder's pastoral production efficiency. The specific model is as follows:

Stage 1: Input-oriented VRS model.

$$min[PTE - \varepsilon(e_1^T s^- + e_2^T s^+)]$$

$$s.t. \begin{cases} \sum_{j=1}^{n} \lambda_j X_j + s^- = TE \cdot X_0 \\ \sum_{j=1}^{n} \lambda_j Y_j - s^+ = Y_0 \\ \sum_{j=1}^{n} \lambda_j = 1 \\ \lambda_j \geq 0, s^- \geq 0, s^+ \geq 0 \end{cases} \tag{2}$$

$$TE = PTE \times SE \tag{3}$$

In (Eq. 2) and (Eq. 3), PTE represents the pure technical efficiency of herder pastoral production. SE represents the scale efficiency of pastoral production. TE represents the integrated technical efficiency of herder pastoral farming, i.e., the pastoral production efficiency of herder, which is the product of the pure technical efficiency of herder pastoral production and the scale efficiency of pastoral production [33]. $X_j$ and $Y_j$ represent the input and output vectors, respectively. $S^+$ and $S^-$ represent slack variables. $\lambda$ represents a vector of weights for different herders. $j = 1, 2, 3, \ldots, n$ represents the jth herder. $\varepsilon$ represents non-Archimedean infinitesimals.

In the grassland pastoral areas of Inner Mongolia, China, after the implementation of the GECP, herders changed their feeding methods to "barn feeding" or "grazing+barn feeding" to raise livestock such as cows, sheep, and horses on pastureland; additionally, herders' main source of income is now from the sale of adult livestock, youngstock and pastoral byproducts such as wool, milk and hides. Moreover, the number of animals stocked and the number of animals slaughtered each year are relatively stable, most herders choose to carry out concentrated feed supplementation during the cold season, and some herders also fatten the weaned cubs to make them come out of the barn in advance. The validity of the efficiency results will be affected if the annual household income of herders or the size of household farms is used as a single output indicator for measuring the efficiency of their pastoral production. Thus, drawing on the studies of Qian et al. [36] and Wang and Du [37], herd income and year-end livestock stock are selected as the output variables and the actual operating area of pastureland of herders, labor inputs, and ranching operating expenses as the input variables, and the specific descriptions of the input and output indicators are shown in Table 1.

Stage 2: Unilateral truncated bootstrap model.

$$T\hat{E}_i = \alpha_0 + \alpha_{1i} Policy_i + \sum \alpha_{2i} Control_i + \varepsilon_i \geq 1 \tag{4}$$

**Table 1. Description and descriptive statistical table of the input and output indicators of livestock breeding in 2023.**

| Form | Specific indicator | Indicator description | Mean | Standard deviation |
|---|---|---|---|---|
| Input indicators | Grassland inputs | Contracted grassland area – forbidden grassland area + renting-in grassland area – renting-out grassland area (ha) | 231.41 | 217.43 |
| | Labor inputs | Work days of own labor + work days of hired labor (work days) | 370.88 | 293.09 |
| | Ranching operating expenses | Forage fee + medical and epidemic prevention fee + depreciation of fixed assets + energy fee (ten thousand yuan) | 14.28 | 24.43 |
| Output indicators | herd income | Herding income of herders (ten thousand yuan) | 21.72 | 23.18 |
| | year-end livestock stock | Livestock stock at the end of the year (sheep units) | 407.30 | 390.60 |

Note: According to the Regulations on Protection of Basic Grasslands in the Inner Mongolia Autonomous Region, livestock are converted into sheep units: 1 goat = 0.9 sheep units, 1 cow = 5 sheep units, 1 horse = 6 sheep units, and 1 camel = 7 sheep units.

where $T\hat{E}_i$ denotes the inverse of the breeding efficiency of the ith herder in the Inner Mongolia pastoral area; $Policy_i$ is the core explanatory variable, which denotes the total amount of grassland ecological compensation received by the herders; $Control_i$ is the other influencing factors; $\alpha_0$ is a constant; $\alpha_{1i}$ is the regression coefficient; $\alpha_{2i}$ is the regression coefficient of the other influencing factors; and $\varepsilon_i$ is the random interference term.

Explained variable: Drawing on the study of Zhang et al. [38], the value of the comprehensive technical efficiency of herds measured by the input-oriented VRS model is taken as the pastoral production efficiency of herders.

Core explanatory variables: In this paper, the total amount of grassland ecological compensation in 2023, which is the sum of the grass–animal balance incentive and the amount of the grazing moratorium subsidy, is selected as the core explanatory variable.

Control variables: In accordance with the studies of Wu et al. [39]and Liu et al. [40], the following perspectives are taken: characteristics of the head of the household, household resource endowment, social resource endowment, and location endowment. The following variables were used as control variables: the age, number of years of education, and training in ranching technology of the head of the household, the share of income from ranching, the number of farming machines, whether pasture is rented or owned, the evaluation of socialized services in animal husbandry, whether the head of the household has animal husbandry insurance, and the distance from the local flag (county) government (Table 2).

**3.3.2. Mediation effect model.** To verify whether moderate-scale operation is a mediating variable in the impact path of the GECP on herders' pastoral production efficiency, the mediating effect model test proposed by Wen and Ye [41] was adopted to test the impact mechanism of the GECP on herders' pastoral production efficiency in the following model:

$$TE_i = \alpha_0 + \alpha_1 Policy_i + \sum \alpha_{2i} Control_i + \varepsilon_i \tag{5}$$

$$MScale_i = \beta_0 + \beta_1 Policy_i + \sum \beta_{2i} Control_i + \varepsilon_2 \tag{6}$$

$$TE_i = \gamma_0 + \gamma_1 Policy_i + \gamma_2 AScale_i + \sum \gamma_{3i} Control_i + \varepsilon_3 \tag{7}$$

Because pastoral production efficiency takes values in the range of [0,1], which belong to the truncated data, (Eq. 5) and (Eq. 7) use the Tobit model, and the herders realize the appropriate scale of operation as a dichotomous variable; referring to the research of Shi et al. [42], (Eq. 6) uses the logistic model. In the above equation, $TE_i$ denotes the breeding efficiency of the ith herder in the Inner Mongolia pastoral area; $Policy_i$ denotes the total amount of grassland ecological compensation obtained by the herder; $MScale_i$ is the intermediary variable, which represents whether the herder realizes moderate-scale operation; and $Control_i$ is the control variable. The reasons for the selection are as follows.

The moderate scale of grassland livestock husbandry refers to the scale of breeding that realizes the win–win situation of ecological protection and economic benefits under the premise of protecting grassland ecology; combining the resource carrying capacity, economic returns and management level; and reasonably controlling the number of animals and the mode of production [43,44]. Moreover, grassland is the most important resource in the process of grassland livestock production and is the basis for the allocation of other factors of production; i.e., when the breeding technology is established, the grassland resources owned by the herdsmen determine the scale of livestock production and the amount of factor inputs [31]. Thus, this paper refers to the study by Zhang et al. [45] to project the moderate scale of livestock rearing based on the pasture carrying capacity of the herders. Since herders' herd size in this paper is the stock of livestock at the end of 2023, the pasture carrying capacity obtained from the "Grassland Monitoring Report of Inner Mongolia Autonomous Region in 2023", which contains information on the livestock-carrying capacity of natural grasslands during the

**Table 2. Description and descriptive statistical table of the influencing factors and variables of breeding efficiency.**

| Form | Specific indicator | Indicator description | N | Mean | Standard deviation | Max | Mix |
|---|---|---|---|---|---|---|---|
| Explained variable | Pastoral production efficiency | Value of integrated technical efficiency of pastoral farming in 2023 | 468 | 0.43 | 0.20 | 0.09 | 1 |
| Core explanatory variables | The total amount of compensation | Grass-animal balance incentive+ in 2023 Pasture rest subsidy in 2023 (ten thousand yuan) | 468 | 1.29 | 1.42 | 0.011 | 9.16 |
| Characteristics of the head of the household | The age of the head of the household | Age of household head (years) | 468 | 49.99 | 8.09 | 20 | 85 |
| | the years of education of the head of the household | Years of schooling of household head (years) | 468 | 8.06 | 3.30 | 0 | 17 |
| | Whether the head of the household has been trained in ranching technology | 1 = yes, 0 = no | 468 | 0.79 | 0.40 | 0 | 1 |
| household resource endowment | The size of the household | Total household size | 468 | 3.54 | 1.12 | 1 | 9 |
| | The proportion of income from ranching | Farming income in 2023/ Total income in 2023 (%) | 468 | 0.78 | 0.23 | 0.03 | 1 |
| | The number of herding machines | Number of farming machinery (units) | 468 | 3.61 | 2.59 | 0 | 16 |
| social resource endowment | Evaluation of socialized services in animal husbandry | 1 = None, 2 = Lower, 3 = Average, the 4 = better, 5 = very good | 468 | 3.31 | 1.08 | 1 | 5 |
| | Whether pastureland is rented | 1 = Yes, 0 = No | 468 | 0.81 | 0.29 | 0 | 1 |
| | Whether the head of the household has animal husbandry insurance | 1 = Yes, 0 = No | 468 | 0.59 | 0.49 | 0 | 1 |
| location endowment | Distance to livestock exchange | Distance to nearest livestock trading market (kilometers) | 468 | 61.08 | 50.82 | 0 | 300 |
| | Area dummy variable 1 | 1 = Eastern, 2 = Other | 468 | 0.26 | 0.44 | 0 | 1 |
| | Area dummy variable 2 | 1 = Central, 2 = Other | 468 | 0.39 | 0.47 | 0 | 1 |

cold season for various herding flags and counties (Table 3). The formula for calculating the herd size and the degree of deviation of herders from the moderate herd size is as follows:

$$Scale = \frac{Grass}{Scc}$$

(8)

$$OR = \frac{(AScale - Scale)}{Scale}$$

(9)

(Eq. 8) is the formula for calculating the moderate herd size of herders, where $Scale$ is the moderate herd size, $Grass$ is the actual area of pasture operated by herdsmen (area of own grassland – area of rented out grassland + area of rented in grassland), and $Scc$ is the capacity of carrying livestock in winter (Table 3). (Eq. 9) is the formula for calculating the degree of herders' deviation from the moderate herd size, where $OR$ is the degree of herders' deviation from the moderate herd size, and $AScale$ is the herders' actual herd size, measured by livestock stock at the end of 2023. In the actual management process of herders, because of the indivisibility of livestock, herders' herd size is within a certain range of dynamic fluctuation states. This paper draws on the research results of Shi et al. [42], in which herdsmen realize moderate-scale operation when $-5\% \leq OR \leq 5\%$, at which time, the mediator variable $MScale = 1$; otherwise, $MScale = 0$. Under the impetus of the GECP, approximately 23.07% of herders realize moderate-scale operation.

**Table 3. Investigation of the winter carrying capacity of natural pastures in the survey area.**

| Area | Union City | Pastoral Flag County | Winter Livestock Carrying Capacity (mu/sheep unit) |
|---|---|---|---|
| Eastern | Hulunbeier City | Xinbaerhuyue Banner | 24.46 |
| | Chifeng City | Arukolqin Banner | 20.46 |
| | | Wengniute Banner | 19.45 |
| Central | Xilingol League | Abaga Banner | 29.68 |
| | | Sunit Right Banner | 45.84 |
| | | West Urumqin Banner | 26.60 |
| Western | Ordos City | Ertokqqi | 40.33 |
| | | Hangjin Banner | 45.87 |
| | | Wutian Banner | 39.02 |

Note: These data come from the Grassland Monitoring Report of the Inner Mongolia Autonomous Region 2023

In this study, we test the mediating effect of herders' realization of moderate-scale operation on the impact mechanism of the GECP on their pastoral production efficiency according to the following five steps: The first step is to test whether the core explanatory variable $Policy_i$ in (Eq. 5) is significant. If it is significant, the argument is based on the mediating effect. Otherwise, the argument is based on the masking effect. However, regardless of whether the difference is significant, a follow-up test is needed. The second step is to test the core explanatory variable $Policy_i$ in (Eq. 6) and the mediating variable $MScale_i$ in (Eq. 7); if both variables are significant, then the indirect effect is significant and the fourth step is performed next. If at least one variable is not significant, then the third step is performed. In the third step, the bootstrap method is used to directly test "$H_0 = \frac{\beta_1}{SE(\beta_1)} \times \frac{\gamma_2}{SE(\gamma_2)}$"; if the original hypothesis is rejected, the indirect effect is significant, and the fourth step is performed. Otherwise, the indirect effect is not significant, and the analysis is stopped. The fourth step is to test whether the core explanatory variable $Policy_i$ in (Eq. 7) is significant; if it is not significant, the direct effect is not significant, and there is only a mediating effect; i.e., there is a complete mediating effect. If it is significant, the direct effect is significant, and the fifth step is performed. In the fifth step, the signs of $\beta_1\gamma_2$ and $\gamma_1$ are compared. If the sign is the same, there is a partial mediation effect, and the mediation effect is $ACEM = \frac{\beta_1}{SE(\beta_1)} \times \frac{\gamma_2}{SE(\gamma_2)}$. If the sign is different, there is a masking effect, and the ratio of the indirect effect to the direct effect is $\left|\frac{ACEM \times SE(\gamma_1)}{\gamma_1}\right|$.

## 4. Results

### 4.1. Analysis of variable returns to scale (VRS) model results

In this study, the rDEA package in the R programming language (4.4.1) was used to measure the pastoral production efficiency of herders and analyze the mechanism of its influence. One hundred interactions were conducted in the first stage of the two-stage semiparametric DEA model, and 2,000 interactions were conducted in the second stage based on Algorithm 2.

The results revealed that compared with the original efficiency, the integrated technical efficiency, pure technical efficiency and scale efficiency of pastoral household farming biased by the two-stage semi-parametric DEA model showed different degrees of decline; similar situations were reported by Linh et al. [46] in measuring the production efficiency of plantation farms in Vietnam and Anang et al. [47] in measuring the maize production efficiency of Ghanaian farm households, as shown in Table 4 and Fig 2. Combined technical efficiency is the product of pure technical efficiency and scale efficiency, and it reflects the overall efficiency of a pastoralist household, considering both technical management and herd size at the current level of technology. The mean value of the bias-corrected integrated technical efficiency value is 0.43, and only 8% of the herders are in the high efficiency range, which indicates that the herders' inputs need to be reduced by approximately 57% to increase the integrated technical efficiency value to 1. Moreover, the lower and upper mean values

**Table 4. Efficiency measurement results.**

|  | Conventional | Corrected | Bais | Lower limit | Higher limit |
|---|---|---|---|---|---|
| TE[a] | 0.48 | 0.43 | −0.06 | 0.41 | 0.45 |
| PTE[b] | 0.53 | 0.49 | −0.13 | 0.42 | 0.52 |
| SE[b] | 0.92 | 0.86 | −0.12 | 0.81 | 0.94 |

[a]TE represents technical efficiency.

[b]PTE represents pure technical efficiency.

[c]SE represents scale efficiency.

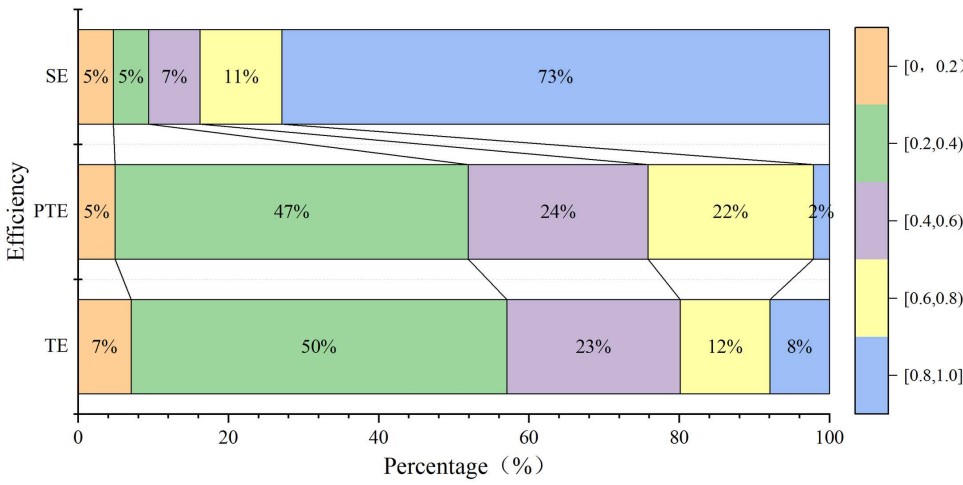

**Fig 2. Distribution of Pastoral Production Efficiency Scores.**

of the bias-corrected integrated technical efficiency value are 0.41 and 0.45, respectively, in the 95% confidence interval, which indicates that an average herder can save approximately 55%–59% of the total inputs by improving the integrated technical efficiency. The pure technical efficiency of herders' farming is the farming efficiency of herders under the existing technical level and management ability after excluding the influence of the scale factor; after calculation, the mean value of the pure technical efficiency value corrected by the herders is 0.49, and only 2% of herders are in the high-efficiency interval, which indicates that herders can still improve their pure technical efficiency by 51% by adopting advanced farming technology and management. Moreover, the pure technical efficiency value corrected by the deviation is 0.49 within the 95% confidence interval, and its lower limit and upper limit mean values are 0.42 and 0.52, respectively. The herd size efficiency reflects the gap between the existing herd size and the theoretical optimal herd size under the existing resource endowment and management ability. The average value of the size efficiency of herders after corrective bias is 0.86, and approximately 78% of herders are in the high-efficiency interval, while the size efficiency value after bias is within the 95% confidence interval. The lower and upper limits of the average value are 0.81 and 0.94, respectively, which indicates that the GECP plays an important role in helping herders resolve the mismatch between past herd size and grassland productivity and apply the "grass for livestock" principle.

## 4.2. Single truncated bootstrap regression analysis

Before regressing the unilateral truncated bootstrap model, the variables were first tested for multicollinearity using the variance inflation factor test, which revealed that the largest variance inflation factor was 1.08 and that the average

variance inflation factor was 1.04, which is much less than 10; thus, there is no serious multicollinearity problem. The core explanatory variable, the total amount of grassland ecological compensation, has a significant positive effect on the pastoral production efficiency of herders in Inner Mongolia pastoral areas at the 1% level (Table 5), and Hypothesis H1 is verified. This finding is consistent with the findings of Ma et al. [29] and Wang et al. [48], who concluded that, on the one hand, the GECP will prompt herdsmen to change their traditional farming methods from free-ranging to "barn feeding or semi confinement" and to improve their farming technology and management ability to adapt to the new farming methods, thus improving pastoral production efficiency. On the other hand, the GECP will constrain the scale of herders' farming through the stipulation of the amount of livestock carrying capacity, and overloaded herders will optimize the structure of the livestock population during the process of livestock reduction; eliminate old, weak and sick animals; and actively introduce good livestock breeds for improvement to increase the production capacity of each unit of livestock, thus improving pastoral production efficiency. Moreover, after receiving the grassland ecological bonuses, the herders will improve the standardized and mechanized level of breeding by improving barn conditions and by purchasing additional breeding machinery, etc., and ultimately improve pastoral production efficiency.

Among the control variables, in terms of the characteristics of the head of the household, the number of years of education of the head of the household has a significant positive effect on the farming efficiency of herders at the 10% level, indicating that the higher the education level of the herders is, the greater their ability to learn and acquire information and to master advanced farming technology to rapidly improve pastoral production efficiency. Whether the head of the household receives technical training has a significant positive effect on the pastoral production efficiency of herders at the 5% level. Currently, most herders in the pasture area apply the farming method of "grazing + barn feeding", so mastering the corresponding foddering farming skills is crucial for herders to improve their pastoral production efficiency. From the point of view of household resource endowment, the proportion of pastoral income has a significant positive effect on the pastoral production efficiency of herders at the 1% level, which may be because the greater the share of farming income in the household income of herders is, the greater the herders' dependence on farming and the greater the degree of importance they attach to farming, which makes it more likely that they can improve their pastoral production efficiency by adopting advanced farming technology and increasing farming investment. The number of pastoral machines has a

**Table 5. Regression results of the unilateral stage bootstrap model.**

| Variable | Regression | 90% confidence interval | 95% confidence interval | 99% confidence interval |
|---|---|---|---|---|
| Bonus amount | −0.2204*** | [-0.3604,-0.0743] | [-0.4674,-0.0860] | [-0.4382,-0.0562] |
| Age of household | −0.0010 | [-0.2999,0.02570] | [-0.0312,0.0305] | [-0.0382,0.0418] |
| Years of education of household | −0.1139* | [-0.1831,-0.0306] | [-0.1859,0.0157] | [-0.2210,0.0434] |
| Whether household head receives technical training | −0.1448** | [-0.3535,-0.2045] | [-0.3853,-0.0705] | [-0.5111,0.2488] |
| Size of family | −0.2558 | [-0.4148,0.0556] | [-0.4356,0.1486] | [-0.5080,0.2028] |
| Percentage of income from herding | −3.7967*** | [-4.6801,-2.8148] | [-4.9177,-2.5394] | [-0.5472,-2.3379] |
| Number of herding machines | −0.1144** | [-0.2195,-0.0122] | [-0.2535,-0.0302] | [-0.2360,0.0520] |
| Evaluation of herding socialization services | −0.5193** | [-0.7005,-0.3508] | [-0.7766,-0.2851] | [-0.8194,0.0903] |
| Whether pasture is rented | −1.0919** | [-1.7853,-0.4566] | [-1.7866,-0.4398] | [-2.0367,0.2341] |
| Whether pasture is insured | −0.2815 | [-0.7773,0.2332] | [-0.7893,0.2581] | [-0.7913,0.5032] |
| Distance to livestock market | 0.0021 | [-0.0014,0.0060] | [-0.0025,0.0075] | [-0.0029,0.5032] |
| Regional dummy variable 1 | −0.1286 | [-0.3148,0.0556] | [-0.3546,0.1538] | [-0.4148,0.1918] |
| Regional dummy variable 2 | 0.2568 | [-0.0148,0.3532] | [-0.1325,0.3968] | [-0.1958,0.4015] |
| Constant term | 9.2715 | [7.0269,11.1200] | [6.8858,11.2289] | [6.5883,12.2265] |

Note: A variable is significant at a certain confidence level if its regression coefficient does not include zero in the confidence interval at that level; the dependent variable for this regression is the inverse of the farming efficiency of the herdsmen.

significant positive effect on the pastoral production efficiency of herders at the 1% level. The use of mechanization is an important indicator of the level of modernization of the pastoral industry. In the pastoral area, each herding household raises an average of 400 sheep units, and the use of pastoral machinery, such as mixers, shears, and haying machines, can greatly reduce labor inputs and improve pastoral production efficiency. From the point of view of social resource endowment, the evaluation of pastoral socialized services has a significant positive effect on the pastoral production efficiency of herders at the 5% level; that is, a higher evaluation of local pastoral socialized services by herders indicates a higher degree of local pastoral socialized services, and the support provided by pastoral socialization service institutions in terms of disease prevention and control guidance, market information and sales channels is relatively comprehensive, which is conducive to reducing the breeding risk and increasing the pastoral production efficiency of herders. Whether the pasture is rented has a significant positive effect on pastoralists' farming efficiency at the 5% level; on the one hand, the GECP directly increases the income of herdsmen through subsidies, causing some herdsmen to quit farming and move to the city. On the other hand, herders who rent in pastureland can realize large-scale farming, which not only reduces live-stock breeding and marketing costs but also facilitates herder introduction of more advanced pasture machinery, breeding techniques and management tools and ultimately improves pastoral production efficiency.

### 4.3. Endogeneity treatment and robustness test

**4.3.1. Endogeneity treatment.** To correct the differences among samples and address the endogenetic issues caused by sample selection bias, in this study, the Propensity Score Matching (PSM) model is used to conduct an in-depth calculation and discuss the impact of the GECP on the pastoral production efficiency of herders.

When this method is employed, it is essential to divide the sample into treatment and control groups. However, the implementation of the Grassland Ecosystem Conservation Project (GECP) occurred simultaneously for all herders in China's major pastoral areas, thus precluding the existence of a natural control group. Consequently, in this study, existing research is used to select samples that are minimally impacted by the policy, thereby constructing a control group [49]. Specifically, herders with less than 100 hectares of contracted pastureland are included in the control group. This selection is based on the understanding that, within the study area, 100 hectares is typically considered the minimum pasture area necessary for a herder to sustain a basic livelihood. Field studies indicate that herders with pasture areas below this threshold tend to make relatively stable livestock production decisions due to livelihood pressures and are less affected, or even negligibly influenced, by policies, rendering them theoretically suitable for inclusion as a control group; i.e., the farming behavior of these herders is unlikely to be significantly influenced by the policy, making it theoretically viable to construct a control group. The rationale for using the contracted pasture area as the basis for grouping is as follows: The contracted pasture area for herders was determined exogenously by local governments in the 1980s. Consequently, this grouping variable is not associated with endogenous factors, such as grazing ability or risk tolerance, that may impact herders' farming efficiency. Therefore, utilizing the contracted pasture area as a grouping variable theoretically preserves the unbiasedness of the estimation results derived from the propensity score matching model.

Based on the aforementioned analysis, this paper employs Stata 17.0 to estimate propensity score matching for the baseline regression model. Table 6 presents the results of pastoral production efficiency and the average treatment effect on the treated (ATT) between the experimental and control groups, obtained through nearest neighbor matching (k = 4), radius matching (0.01), and kernel matching. The results indicate that the estimates derived from these three matching methods are largely consistent, suggesting that the findings are robust. The average value of ATT is approximately 0.0702, which implies that the implementation of the GECP increases the pastoral production efficiency of herders by 0.0702 after controlling for confounding variables. These findings indicate that the GECP positively influences the breeding efficiency of herdsmen. The common support domain serves as a fundamental test objective for assessing the efficacy of the Propensity Score Matching (PSM) method. To ensure the validity of the matching results, it is essential to examine the common support domain of both the treatment and control groups. By analyzing the trend of the propensity

**Table 6. The results of PSM.**

| Matching Methods | ATT | S.E. |
|---|---|---|
| Nearest Neighbor Matching (K=4) | 0.0799** | 0.0349 |
| Radius Matching (0.01) | 0.0693** | 0.0344 |
| Kernel Matching | 0.0614* | 0.0331 |
| Mean Value | 0.0702 | – |

*** refers to the significance level of 1%.

score values for samples in both groups after applying the three matching methods, we can identify the overlap, which constitutes the common support domain (Fig 3). The findings reveal that most samples in both groups fall within the same range, with only a minimal number of samples being excluded after propensity score matching. This outcome increases the robustness of the common support domain in this study.

Finally, after matching, a balancing test was used to determine whether the variables were balanced between the control and treatment groups. The results of the balancing test are shown in Table 7. The Pseudo R2 value decreased from 0.094 before matching to 0.004~0.012. The LR statistic decreased from 39.07 to 0.90~2.90. The probability value changed from significant to insignificant. The mean deviation decreased from 19.5% to 3.30~6.20%. The median deviation decreased from 20.70% to 2.8~5.60%. In summary, the total error of the matched samples was greatly reduced, and the characteristics of the samples were similar between groups. The balance test was passed.

**4.3.2. Robustness test.** To test the robustness of the results of the two-stage semi-parametric DEA model, the traditional two-stage model (DEA+Tobit) from the study of Long et al. [50] is applied; in this model, the dependent variable is the value of the original pastoral production efficiency measured by the DEA model, and the regression coefficients of the explanatory variables are positive, which indicates that the variable has a positive influence on the pastoral production

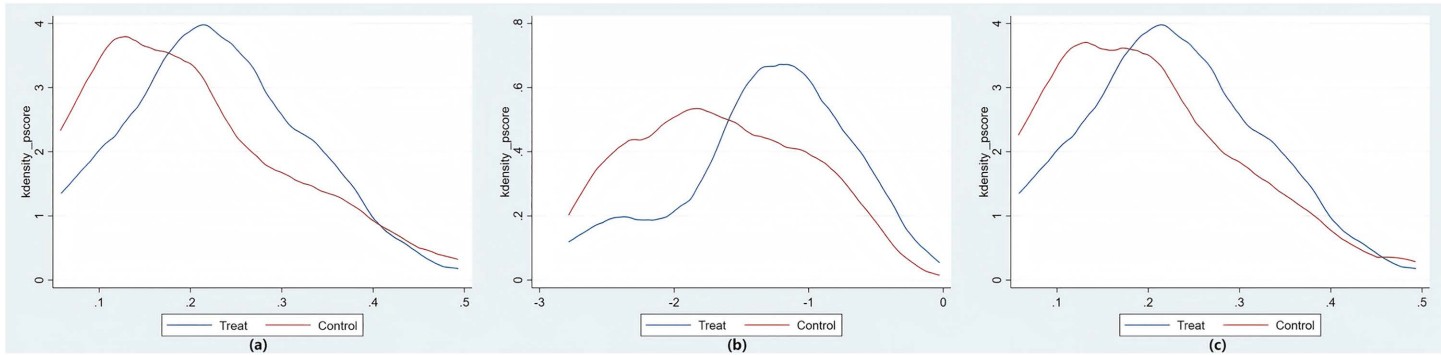

**Fig 3. Kernel Probability Distribution after Propensity Score Matching.**

**Table 7. The results of the balance test after variable matching.**

| Matching Methods | Pseudo R² | LR Statistic | p Value | Mean Deviation | Median Deviation |
|---|---|---|---|---|---|
| Before Matching | 0.094 | 39.07 | 0.000 | 19.5 | 20.7 |
| Nearest Neighbor Matching (K=4) | 0.012 | 2.62 | 0.998 | 6.2 | 4.8 |
| Radius Matching (0.01) | 0.014 | 2.90 | 0.996 | 6.6 | 5.6 |
| Kernel Matching | 0.004 | 0.90 | 1.000 | 3.3 | 2.8 |

efficiency of the herder. The regression results of the Tobit model are shown in Table 8, and it can be seen that the direction of the influence of the explanatory variables on the dependent variable of this model and the two-stage semi-parametric DEA model are consistent, and the significance level of only two variables are slightly different: the evaluation of pasture socialized services and whether pasture is rented in. Thus, the conclusions presented in Table 5 are more reliable.

## 4.4. Validation of mediation effect models

In this paper, according to the mediating effect model test method, the overall fitting effect of the mediating effect model is better (Table 9). The model results of (Eq. 5) show that the total amount of the GECP ecological bonus has a significant positive effect on the breeding efficiency of the herder at the 1% level; i.e., the core explanatory variable of the total amount of grassland ecological subsidies is significant, which indicates that there is a mediating effect. The results of the model in Eq. (6) show that the total amount of grassland ecological bonus has a significant positive effect on whether the herder realizes moderate-scale management at the 5% level; i.e., the mediating variable in Eq. (6) is significant, as is the herder's realization of moderate-scale management. The results of the model in Eq. (7) show that the herder's realization of moderate-scale management has a significant positive effect on their pastoral production efficiency at the 10% level, which means that there are mediating effects and the mediator variable in Eq. (7) is significant, which indicates that the indirect effect is significant. After the mediator variable of herders' realization of moderate-scale management is added to the model of Eq. (7), the core explanatory variable total grassland ecological subsidy is also significant, which indicates that the direct effect is significant. In the above regression results, both are positive; that is, the indirect effect of herders' realization of moderate-scale management on their pastoral production efficiency is in the same direction as the direct effect, indicating that there is a part of a mediating effect and that the mediating effect accounts for 18.42% of the total effect. Therefore, the GECP has the mechanism of 'GECP implementation – herder realization of appropriate scale management – improved pastoral production efficiency', and Hypothesis H2 is verified. In the past, due to the constraints

**Table 8. Tobit model regression results table.**

| variable | coefficient of regression | Standard error | t value | p value |
|---|---|---|---|---|
| Total amount of supplementary prize | 0.0188*** | 0.0070 | 2.639 | 0.0083 |
| Age of head of household | 0.0006 | 0.0012 | 0.539 | 0.5898 |
| Number of years of education of the head of household | 0.0052* | 0.0029 | 1.785 | 0.0743 |
| Does the householder receive technical training? | 0.0279** | 0.0118 | 2.360 | 0.0183 |
| Family size | 0.0078 | 0.0086 | 0.902 | 0.3671 |
| Proportion of animal husbandry income | 0.1982*** | 0.0434 | 4.546 | 0.0005 |
| Quantity of animal husbandry machinery | 0.0088** | 0.0038 | 2.359 | 0.0183 |
| Evaluation of socialized services of animal husbandry | 0.0247*** | 0.0091 | 2.679 | 0.0070 |
| Whether pastureland is rented | 0.0937*** | 0.0248 | 3.779 | 0.0002 |
| Whether the head of the household has animal husbandry insurance | 0.0185 | 0.0199 | 0.927 | 0.3541 |
| Distance from livestock trading market | −0.0001 | 0.0002 | −0.250 | 0.8024 |
| Region virtual variable 1 | 0.0433 | 0.1221 | 1.951 | 0.1510 |
| Regional virtual variable 2 | −0.0437 | 0.0201 | −0.211 | 0.8327 |
| constant term | −0.0347 | 0.0934 | −0.371 | 0.7107 |
| LR chi2(11) | 71.96 | | | |
| Prob>chi2 | 0.0000 | | | |

Note: * indicates the 10% significance level, * * indicates the 5% significance level and * * * indicates the 1% significance level; the same is true below.

of the scale of grassland animal husbandry and the degree of specialization of herders, herd size exceeded the capacity of grassland ecology, which led to the deterioration of grassland ecology, a slow increase in herder income, low economic efficiency of grassland animal husbandry, and other problems. Through the establishment of ecological bonuses and strengthening supervision, herders were encouraged to reduce herd size so that herd size and grassland production capacity match to avoid uneconomical scaling. In this process, herders will consider their own grassland carrying capacity to readjust the input of resources and optimize the allocation of resources to achieve the appropriate scale of operation, the formation of a pasture restoration. A positive cycle of 'pasture restoration – pasture quality improvement – increased livestock efficiency' is formed. Linh et al. [46] reported that when the actual livestock loading rate of herders was reduced to the optimal loading rate, although the total expenditure and total income of herders decreased, their net income and net income per unit of livestock increased. Thus, pastoralists improved their pastoral production efficiency by achieving an appropriate scale of operation.

## 4.5. Heterogeneity analysis

Grassland resources, which are the most important production factors in the farming activities of herders, are not only the approved criteria for the release of ecological compensation but also the core variable for calculating the theoretical livestock carrying capacity. According to the theory of resource endowment, in the context of the implementation of the GECP, pasture resources directly determine the adjustment space of the production behavior of herders and the potential for improvement in pastoral production efficiency. Therefore, to analyze the heterogeneous impact of the GECP on the pastoral production efficiency of herders, the sample is divided into a large-herding-household sample (the actual area of pasture operated by herders is larger than the average) and a small-herding-household sample (the actual area of pasture operated by herders is smaller than the average) according to the actual area of pasture operated by herders in 2023, and a mediating effect model is established to analyze the two samples separately. The results are shown in Table 10.

It was found that the impact of the GECP on the pastoral production efficiency of herders was still dominated by direct effects, both in the sample of large-scale herders and in the sample of small-scale herders, and that the direction of the mediation transmission of the moderate-scale operation was in the same direction as the direct effect, which was manifested as a partially mediated effect. These findings are largely consistent with the results of the overall sample. Further analysis revealed that the indirect effect of the GECP on the pastoral production efficiency of small-scale herders (22.34%) is significantly greater than that of large-scale herders (11.78%), which may be because small-scale herders used to be the main body of overloaded grassland pasture areas, and the resource allocation of these herders is imbalanced. The GECP encourages small-scale herders to reduce the scale of their breeding by means of supervision and subsidies and guides them to control the herd size considering the carrying capacity of the pasture, resolving the previous vicious circle of 'pasture degradation – overgrazing' and indirectly improving pastoral production efficiency by realizing the appropriate scale of operation. Comparatively speaking, large-scale herders have richer pasture resources, and these

**Table 9. Results of the Mediation Effect Model.**

| model | Tobit model | Logistic model | Tobit model |
|---|---|---|---|
| variable | TE | Mscale | TE |
| Policy | 0.0380*** (0.006) | 0.3641** (0.074) | 0.0310*** (0.007) |
| Mscale | -- | -- | 0.0418* (0.023) |
| Control variable | Controlled | Controlled | Controlled |
| Pseudo R²/R² | 0.168 | 0.151 | 0.173 |

**Table 10. The results of the heterogeneous herders' mediating effect model.**

| Herder Type | Model Variable | Tobit model TE | Logistic model MScale | Tobit model TE |
|---|---|---|---|---|
| large-scale herders | Policy | 0.0713*** (0.011) | 0.3180** (0.115) | 0.0629*** (0.010) |
| | Mscale | -- | -- | 0.1137*** (0.032) |
| | Control variable | Controlled | Controlled | Controlled |
| | Pseudo R²/R² | 0.220 | 0.039 | 0.271 |
| | Sample size | 161 | | |
| small-scale herders | Policy | 0.0613* (0.027) | 0.4238*** (0.185) | 0.0571* (0.028) |
| | Mscale | -- | -- | 0.1418** (0.031) |
| | Control variable | Controlled | Controlled | Controlled |
| | Pseudo R²/R² | 0.113 | 0.052 | 0.118 |
| | Sample size | 307 | | |

herders are better than small-scale herders in terms of both breeding scale and the total amount of subsidies received; thus, they will increase their investment in breeding, such as sheltered sheds and breeding machinery, and thus improve their breeding technology and management level, and the direct effect of the GECP on pastoral production efficiency is even greater. This pattern of differentiation suggests that policy design needs to consider the characteristics of the group and that small-scale herders should work together to strengthen the supervision of the grass–animal balance and support alternative livelihoods, whereas large-scale herders need to guide the flow of compensatory funds to technological innovation to ultimately realize the scale-matched development path of ecological protection and the transformation of the pastoral industry.

## 5. Discussion

Based on 468 field research questionnaires in pastoral areas of Inner Mongolia, in this paper, the effect of the GECP on the pastoral production efficiency of herders and the influence mechanism on the pastoral production efficiency of herders are empirically analyzed, and the mechanism of herders' realization of the role of the appropriate scale of operation in the relationship between the two is clarified. The results revealed that in the pastoral areas of Inner Mongolia, China, although the GECP has improved the pastoral production efficiency of herders, the policy effect is more limited, and the current breeding efficiency of herders in this pastoral area as a whole is still in the lower efficiency range, mainly because of the lower pure technical efficiency, and pastoral production efficiency still has much room for improvement, which is consistent with the conclusions of some previous studies [34,47,48]. The incentive effect of the implementation of the GECP on the development of the pasture industry is due to the standardization of the use of pasture, guiding herders to change to the modernized farming method of "grazing + barn feeding" while compensating for the production costs incurred during the early stage of the transformation so that the herdsmen adapt to the new way of farming and learn advanced farming techniques and management methods to directly improve their farming efficiency. However, it cannot be ignored that the pastoral areas of Inner Mongolia are sparsely populated and face the same "aging population" dilemma as China's agricultural areas [33], resulting in the slow diffusion and adoption of farming management techniques, which seriously restricts the economic development of pastoral areas.

Through a mechanism test, this paper revealed that the realization by herders of moderate-scale operations in the GECP affects herders' pastoral production efficiency and that the indirect effect of the policy on small-scale herders is significantly greater than that on large-scale herders. In the past, the grass–animal relationship in Inner Mongolia pastoral

areas was not always imbalanced, and its evolution can be divided into four major stages based on the influence of climate and policy: Stage 1: the pasture surplus stage of "grass> livestock" before 1965; Stage 2: "Grass ≈ Livestock" stage during 1966–1985; Stage 3: "Livestock> Grass" imbalance stage during 1986–2005; and Stage 4: "Livestock> Grass" imbalance recovery stage after 2006 [17]. In pastoral areas, the dynamic balance of grass and livestock is the basis for high-efficiency farming. With increasing ecological compensation and supervision, the imbalance of "livestock>grass" is being recovered at a faster pace, and herders are gradually realizing appropriate-scale operations and optimizing resource allocation, thereby indirectly improving their pastoral production efficiency. Moreover, compared with large-scale herders, small-scale herders are more likely to fall into the serious imbalance of "livestock> grass" [49]; under the influence of the GECP, herders gradually transfer part of the family labor force to other industries and reduce the scale of breeding to achieve the optimal allocation of breeding resources with the core of the pasture such that the policy has a positive impact on small-scale herds' pastoral production efficiency, which is more prominent in the indirect impact of the policy.

The key to achieving the policy objective of synergistic development of ecology, production and life in grassland pastoral areas and the dynamic balance of "people, grass and animals" lies in improving pastoral production efficiency and accelerating the transformation and improvement of grassland animal husbandry, thus improving the economic benefits to herders. Only when herders achieve sustainable livelihoods will it be easier to motivate them to protect grassland ecology. Research data show that 67.30% of herders are willing to take the initiative to contribute to the protection of grassland ecology. When the grassland ecology is gradually restored, productivity will continue to increase, the development of the pastoral industry will accelerate, and a virtuous cycle of "pastoral development – increased herder income – ecological restoration" will ultimately be realized (Fig. 4). How to ensure that this cycle operates more efficiently is the direction for optimizing the fourth round of grassland ecological bonus policies. Owing to the complexities of the research, this study exclusively utilized cross-sectional data from the questionnaire survey. However, such data present certain limitations. First, the sparse population distribution in pastoral areas makes it challenging to obtain comprehensive survey data. This study includes only 468 sample herders from four leagues and cities in Inner Mongolia, resulting in a relatively limited sample size. Consequently, the empirical results may not fully represent the actual conditions of the entire pastoral area, and their reliability requires further verification. Second, cross-sectional data fail to capture the dynamic characteristics of variables over time and cannot reflect the longitudinal evolution of herders' behaviors and production efficiency. Therefore, the results of this paper cannot be used to make conclusions about the dynamic correlation and causal mechanisms between the GECP and pastoral production efficiency. Future research investigating the relationship between this policy and pastoral production efficiency should consider including additional representative pastoral areas in China, such as Qinghai, Tibet, and Xinjiang. By broadening the data coverage and increasing data quality, the representativeness and explanatory power of the research can be significantly improved.

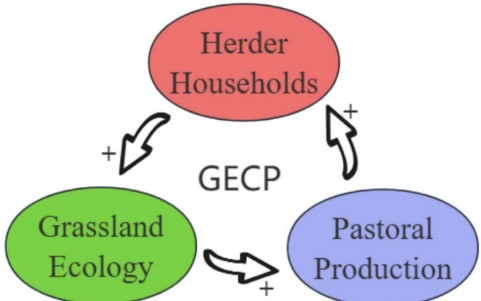

**Fig 4. Multidimensional Effects of GECP Implementation.**

## 6. Conclusions

This paper is based on data from 468 questionnaires completed by herders in the pastoral areas of Inner Mongolia, namely, Hulunbeier city, Xilingol League, Chifeng city and Ordos city, during field research in the implementation period of the third round of the GECP. First, the two-stage semi-parametric DEA model was used to measure the integrated technical efficiency, pure technical efficiency and pastoral production efficiency of the herders' farming; then, the integrated technical efficiency was used as a representative of the herder's pastoral production efficiency. The integrated technical efficiency was used to represent the herders' pastoral production efficiency. The comprehensive technical efficiency values were used to represent the pastoral production efficiency of the herders, and then the impact effect and impact mechanism of the GECP on the pastoral production efficiency of the herders were investigated separately. Finally, whether the moderate-scale operation was realized was taken as the mediator variable, and the mediator effect model was used to verify the impact mechanism of the GECP on the pastoral production efficiency of the herders and to obtain the following conclusions:

(1) The pastoral production efficiency level of herders in the pastoral areas of Inner Mongolia, China, is low overall, with a mean value of 0.43, and there is an efficiency loss of 57%, leaving much room for improvement. The mean value of the pure technical efficiency of farming is 0.49, and the mean value of the scale efficiency is 0.86; the lower level of pastoral production efficiency is due mainly to the lower level of the pure technical efficiency of farming.

(2) The GECP significantly promotes the improvement of herders' pastoral production efficiency, and among the control variables, the number of years of education of the head of household, whether the head of household receives technical training, the proportion of income from herding, the number of herding machines, the evaluation of herding socialized services and whether the pasture is rented have a significant positive effect on herders' pastoral production efficiency.

(3) Herders' realization of an appropriate scale of management of herd size plays a part in the mediating the impact mechanism of the GECP on herder's pastoral production efficiency, with the mediating effect accounting for 18.42%. In addition, the results of the heterogeneity analysis revealed that the direct effect of the GECP on the pastoral production efficiency of large-scale herders was significantly greater than that of small-scale herders.

## 7. Recommendations

(1) Promote advanced breeding techniques to increase the efficiency of pastoral production. The government should enhance the socialized service system for animal husbandry and strengthen the relevance and accessibility of technology promotion to improve the generally low breeding efficiency observed among pastoral households. Concurrently, it is essential to actively conduct pastoral technology training to increase the overall quality of herdsmen, thereby aligning their skills with the demands of modern animal husbandry and increasing pastoral production efficiency.

(2) Continuously improve the grassland ecological reward policy and its supporting policies. It is imperative to continuously refine the supporting policies of the GECP and facilitate the transformation and improvement of grassland animal husbandry. Governments at all levels must bolster the construction of basic security capabilities and increase policy support for essential components such as shed construction, high-quality forage base development, and the procurement of breeding machinery. These efforts should be systematically integrated into the framework of the ecological subsidy award policy to alleviate the capital constraints and technological bottlenecks that herders encounter during the initial stages of breeding method transformation. By increasing the control-lability of herders in shelter-feeding breeding, we can effectively improve their breeding conditions, thereby promoting the successful transformation and improvement of grassland animal husbandry.

(3) Guide herders to adopt appropriate scale management strategies. The government should establish differentiated standards for moderate-scale operations that align with the needs of herdsmen and the ecological conditions of grasslands. Furthermore, it is imperative to promote the concept of sustainable animal husbandry among herdsmen. Encouraging the formation of professional cooperatives, family ranches, and other organizational structures will facilitate the integration and optimization of production factors, including pasture, machinery, and labor. The aim of this approach is to increase resource allocation efficiency and ultimately improve breeding technology and enhance economic benefits.

## Supporting information

**S1 File. Data.**
(XLSX)

**S2 File. Code.**
(DOCX)

**S3 File. Propensity score matching method code.**
(DOCX)

## Author contributions

**Conceptualization:** Fang Ju, Wenjie Ouyang, Chengtao Zhang, jianjun zhang.

**Data curation:** Fang Ju, Wenjie Ouyang, Chengtao Zhang, jianjun zhang.

**Formal analysis:** Fang Ju, Wenjie Ouyang, Chengtao Zhang.

**Funding acquisition:** Fang Ju.

**Investigation:** Fang Ju, jianjun zhang.

**Methodology:** Fang Ju, Wenjie Ouyang, Chengtao Zhang.

**Project administration:** Fang Ju, jianjun zhang.

**Software:** Wenjie Ouyang, Chengtao Zhang.

**Supervision:** jianjun zhang.

**Validation:** jianjun zhang.

**Visualization:** jianjun zhang.

**Writing – original draft:** Fang Ju, Wenjie Ouyang.

**Writing – review & editing:** Fang Ju, Wenjie Ouyang.

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
