## [Decision Letter · Decision Letter 0]

20 Aug 2025

PLOS ONE

Dear Dr. zhang,

Thank you for submitting your manuscript to PLOS ONE. After careful consideration, we feel that it has merit but does not fully meet PLOS ONE’s publication criteria as it currently stands. Therefore, we invite you to submit a revised version of the manuscript that addresses the points raised during the review process.

This paper addresses a valuable topic with sound methods, but requires minor revisions to solidify its contribution and robustness. Please explicitly clarify the study's novelty in resolving existing literature controversies and directly state its marginal contributions. The most critical revisions involve addressing endogeneity concerns, such as selection bias, potentially via PSM, and providing a strong scholarly justification for the DEA model's use of both flow and stock output variables. Furthermore, the manuscript needs expanded descriptive statistics, a detailed sampling description, a discussion of cross-sectional data limitations, and full definitions for all abbreviations. Finally, a thorough language edit and tighter alignment of policy recommendations with the specific findings are essential for publication.

We look forward to receiving your revised manuscript.

Kind regards,

Abebayehu Aticho (PhD, Associate Prof.)

Academic Editor

PLOS ONE

Journal Requirements:

3. Please provide additional details regarding participant consent. In the ethics statement in the Methods and online submission information, please ensure that you have specified (i) whether consent was informed and (ii) what type you obtained (for instance, written or verbal, and if verbal, how it was documented and witnessed). If your study included minors, state whether you obtained consent from parents or guardians. If the need for consent was waived by the ethics committee, please include this information.

4. We suggest you thoroughly copyedit your manuscript for language usage, spelling, and grammar. If you do not know anyone who can help you do this, you may wish to consider employing a professional scientific editing service.

5. Please note that PLOS One has specific guidelines on code sharing for submissions in which author-generated code underpins the findings in the manuscript. In these cases, we expect all author-generated code to be made available without restrictions upon publication of the work. Please review our guidelines at https://journals.plos.org/plosone/s/materials-and-software-sharing#loc-sharing-code and ensure that your code is shared in a way that follows best practice and facilitates reproducibility and reuse.

6. In the online submission form, you indicated that the original contributions presented in the study are included in the article/Supplementary Material. Further inquiries can be directed to the corresponding authors.

7. Thank you for stating the following in the Acknowledgments Section of your manuscript:

This research was funded by the National Natural Science Foundation of China Project “Study on the Influence Mechanism of Grassland Ecological Compensation and Reward Policy on Part-time behavior of Grazing Herdsmen——Taking Inner Mongolia as an Example” (72363025)

This research was funded by the National Natural Science Foundation of China Project “Study on the Influence Mechanism of Grassland Ecological Compensation and Reward Policy on Part-time behavior of Grazing Herdsmen——Taking Inner Mongolia as an Example” (72363025)

The roles played by funders in research include design, data collection and analysis, and decision-making for publication.

8. Your abstract cannot contain citations. Please only include citations in the body text of the manuscript, and ensure that they remain in ascending numerical order on first mention.

9. Please include a separate caption for each figure in your manuscript.

10. Please upload a copy of Supporting Information Figure 1, 2, 3, which you refer to in your text on page 19.

Reviewers' comments:

Reviewer's Responses to Questions

**Comments to the Author**

1. Is the manuscript technically sound, and do the data support the conclusions?

Reviewer #1: Yes

Reviewer #2: Yes

2. Has the statistical analysis been performed appropriately and rigorously?

Reviewer #1: Yes

Reviewer #2: Yes

3. Have the authors made all data underlying the findings in their manuscript fully available?

Reviewer #1: Yes

Reviewer #2: Yes

4. Is the manuscript presented in an intelligible fashion and written in standard English?

Reviewer #1: Yes

Reviewer #2: Yes

Reviewer #1: The topic of this paper holds significant practical and policy value. The research design is relatively complete, and the use of cutting-edge methods such as the two-stage semi-parametric DEA model and the mediation effect model provides a valuable exploration for understanding the economic effects of the Grassland Ecological Compensation Policy (GECP). The findings that GECP enhances herders' production efficiency and that appropriate-scale management plays a mediating role are of reference value for policy optimization.

1.The paper summarizes existing research on the ecological and economic (income) effects of GECP, but it does not sufficiently review domestic and international studies on the impact of similar ecological compensation policies on production efficiency. The paper notes that research on GECP's effect on production efficiency is scarce and the conclusions are controversial, but it fails to conduct an in-depth exploration of the reasons for these controversies. The paper states, "while some studies have also found that ecological compensation is difficult to compensate for the increase in the cost of shepherding as a result of the policy, and the longer the time of the ban on animal husbandry, the greater the decrease in the efficiency of technological progress of the herding households," but it does not clearly elaborate on how the present study innovates upon or improves existing literature.

2.Although the paper mentions its main research content at the end of the introduction or literature review, its elaboration on the paper's innovativeness or marginal contribution is not sufficiently clear or specific. It is recommended to supplement this further.

3. The paper primarily uses cross-sectional data from a questionnaire survey, which has many limitations for this type of research. Please further elaborate and discuss these limitations.

4. The descriptive statistics for the variables are rather simplistic. Typically, they should include indicators such as the sample size (N), mean, standard deviation, maximum, and minimum values.

5.The paper fails to discuss the potential endogeneity problem. The empirical results show a positive correlation between the ecological compensation policy and efficiency, but this may not be entirely a policy effect. It could partially stem from a selection bias, where "superior" herders (those with higher innate ability or better management skills) are both more efficient and receive more subsidies.

6.In the DEA model's input-output indicators, the paper uses "herding income" (a flow indicator) and "year-end livestock stock" (a stock indicator) simultaneously as output variables. The scientific validity of this approach is questionable, and the authors should provide a sufficient scholarly justification for this choice.

Reviewer #2: This article investigates the impact of the Grassland Ecological Compensation Policy (GECP) on the livestock production efficiency of herders in Inner Mongolia, China. It employs an innovative two-stage semiparametric DEA model and a mediation effect model for analysis. The study design is reasonable, data collection is comprehensive, and the analytical methods are appropriate. The conclusions have significant reference value for policy-making. The article meets the publication standards of PLOS ONE but still has several aspects that need improvement.

1. Abbreviations must be defined at first mention (such as DEA, VRS, etc.).

2. The description of the sample selection process is not detailed enough, and the representativeness of the sampling method needs further explanation.

3. The chart titles and descriptions can be more detailed, and some tables lack necessary annotations. For instance, table 4, what do TE, PTE and SE stand for?

4. The use of terminology needs to be more consistent (such as mixing 'herding households' and 'herders').

5. The policy recommendation section can be more specific and more closely related to research findings.

6. English writing requires further polishing, some sentences have grammatical errors and issues with expression fluency.

**Do you want your identity to be public for this peer review?** For information about this choice, including consent withdrawal, please see our Privacy Policy

Reviewer #1: No

Reviewer #2: No

---

## [Author Response · Author response to Decision Letter 1]

24 Sep 2025

Response to Reviewer 1 Comments

Title Impact of the grassland ecological compensation policy on pastoral production efficiency—evidence from pastoral China

Manuscript ID: PONE-D-25-40838

Thank you for your meticulous revisions of our articles in your busy schedule, which have helped us immensely in improving the quality of our articles. We have revised the manuscript in accordance with the comments and marked all the amends on our revised manuscript. The reviewer comments are laid out below in italicized font and specific concerns have been numbered. Our responses are listed in red font, the specific modified contents are listed in blue font, and the modified contents are highlighted in yellow in the manuscript.

[Comments 1]�The paper summarizes existing research on the ecological and economic (income) effects of GECP, but it does not sufficiently review domestic and international studies on the impact of similar ecological compensation policies on production efficiency. The paper notes that research on GECP's effect on production efficiency is scarce and the conclusions are controversial, but it fails to conduct an in-depth exploration of the reasons for these controversies. The paper states, "while some studies have also found that ecological compensation is difficult to compensate for the increase in the cost of shepherding as a result of the policy, and the longer the time of the ban on animal husbandry, the greater the decrease in the efficiency of technological progress of the herding households," but it does not clearly elaborate on how the present study innovates upon or improves existing literature.

[Response 1]�Thank you very much for your high-quality suggestions, they help us tremendously to improve the quality of our articles.In the literature review section, we supplemented the discussion on the impact of Payment for Ecosystem Services (PES) on production efficiency, deeply analyzed the root causes of the debate, and finally clearly expounded how this study innovates and improves on the basis of existing literature. The specific revised contents can be found in lines 91-106, 133-162 of the revised draft.

[Comments 2]�Although the paper mentions its main research content at the end of the introduction or literature review, its elaboration on the paper's inattentiveness or marginal contribution is not sufficiently clear or specific. It is recommended to supplement this further.

[Response 2]�Thank you for your rigorous and constructive suggestions. We have provided a more detailed and specific elaboration on the innovation or marginal contribution of the paper. The specific revised contents can be found in lines 149-162 of the revised draft.

[Comments 3]�The paper primarily uses cross-sectional data from a questionnaire survey, which has many limitations for this type of research. Please further elaborate and discuss these limitations.

[Response 3]�Thank you for your valuable suggestions. In accordance with your opinions, we have further elaborated in detail on the limitations of cross-sectional data in the discussion section of the paper. Specific changes can be made to lines 789-803 in the revised draft.

[Comments 4]�The descriptive statistics for the variables are rather simplistic. Typically, they should include indicators such as the sample size (N), mean, standard deviation, maximum, and minimum values.

[Response 4]�Thank you for your careful review of this paper. Based on your feedback, we have provided a more detailed description of the variable.

[Comments 5]�The paper fails to discuss the potential endogeneity problem. The empirical results show a positive correlation between the ecological compensation policy and efficiency, but this may not be entirely a policy effect. It could partially stem from a selection bias, where "superior" herders (those with higher innate ability or better management skills) are both more efficient and receive more subsidies.

[Response 5]�Thank you for your valuable suggestions. In response to your feedback, we have incorporated a discussion on endogenous issues within the robustness test section of the article. To mitigate potential sample self-selection bias, we employed the propensity score matching (PSM) method to further validate the benchmark regression results. The specific steps are as follows: First, we categorized the sample herders into a treatment group (those participating in the project) and a control group (those not participating in the project) based on whether the area of household contracted grassland exceeds 100 hectares. Subsequently, we conducted a propensity score matching analysis using Stata 17.0. Finally, we performed an empirical test based on the matched samples to assess the robustness of the estimated results after controlling for selection bias.Specific changes can be made to lines 565-620 in the revised draft.

[Comments 6]�In the DEA model's input-output indicators, the paper uses "herding income" (a flow indicator) and "year-end livestock stock" (a stock indicator) simultaneously as output variables. The scientific validity of this approach is questionable, and the authors should provide a sufficient scholarly justification for this choice.

[Response 6]�Thank you for your rigorous and constructive suggestions. In this paper, we utilize a combination of two types of variables: herd income and year-end livestock stock. This approach is adopted to mitigate the potential bias associated with relying on a single indicator, thereby enhancing the accuracy of our research results. Specifically, herd income, which serves as the "economic flow" from current farming activities, encompasses transfer income from the sale of livestock, wool, and cashmere, as well as grassland ecological subsidies. This variable directly reflects the profitability of production activities during the current period. In contrast, year-end livestock stock represents the "stock of assets" related to farming activities, illustrating the unrealized reserves of production resources and the future output potential of herders.

Measuring the production capacity of herding households solely by "herd income" can lead to results that are easily influenced by external factors, such as fluctuations in market prices. Conversely, relying exclusively on "year-end livestock stock" fails to account for hidden factors, including market conditions, the quality of livestock breeds, and product premiums. The integration of both types of indicators provides a more comprehensive understanding and assists in identifying anomalies, thereby enhancing data quality assessment. For instance, if a herder reports a high inventory yet generates very low income from livestock farming, this discrepancy may indicate data misreporting (e.g., an inflated inventory figure) or special circumstances (e.g., inability to sell livestock due to epidemics). Conversely, if a herder has a low inventory but unusually high income, it is essential to investigate whether this is attributable to exceptional factors, such as the sale of rare breeds or substantial subsidies received.

This method has also been widely employed in existing studies. For instance, Qian (2019) utilized both "livestock stock at the end of the year" and "livestock income of herdsmen" as output indicators when analyzing the moderate scale of operation of herdsmen across different grassland types in Inner Mongolia. Similarly, Wang (2023) employed both "number of livestock slaughtered" and "livestock income" as indicators to examine the impact of operational scale on comprehensive efficiency and operational effectiveness.

In summary, the concurrent use of "livestock income" and "year-end stocking" as output indicators can significantly mitigate the risk of distortion associated with a single indicator, thereby enhancing the reliability and explanatory power of the calculation results.We also explain the selection of these two indicators in the article. Detailed changes can be found in lines 326-368 of the article.

Reference

[1]Qian G X, Zhang N, Qian F M. Study on moderate scale of grassland animal husbandry under different target orientations: Based on herder data from four grassland types in Inner Mongolia[J]. Agricultural Economics and Management, 2019, (02): 55-66.

[2]Wang H, Du F L. Empirical study on the impact of herders' operation scale on comprehensive efficiency and economic benefits: A case study of typical grassland areas[J]. Journal of Agricultural Technical Economics, 2023, (08): 100-112. doi:10.13246/j.cnki.jae.2023.08.008.

Response to Reviewer 2 Comments

Title Impact of the grassland ecological compensation policy on pastoral production efficiency—evidence from pastoral China

Manuscript ID: PONE-D-25-40838

Thank you for your meticulous revisions of our articles in your busy schedule, which have helped us immensely in improving the quality of our articles. We have revised the manuscript in accordance with the comments and marked all the amends on our revised manuscript. The reviewer comments are laid out below in italicized font and specific concerns have been numbered. Our responses are listed in red font, the specific modified contents are listed in blue font, and the modified contents are highlighted in yellow in the manuscript.

[Comments 1]�Abbreviations must be defined at first mention (such as DEA, VRS, etc.).

[Response 1]: Many thanks for your valuable suggestions, which have been extremely helpful in improving the overall quality of our manuscript. In response,We provided a detailed definition when the abbreviation was first mentioned.The full name of DEA is "Data Envelopment Anlysis".The full name of CRS is "Constant Returns to Scale". The full name of VRS is "Variable Returns to Scale".

[Comments 2]�The description of the sample selection process is not detailed enough, and the representativeness of the sampling method needs further explanation.

[Response 2]: We greatly appreciate the insightful changes you have suggested. In response, We have provided a more detailed description of the sample selection process and a more specific explanation of the representativeness of the sampling method. Specific changes can be made to lines 231-260 in the revised draft.

[Comments 3]�The chart titles and descriptions can be more detailed, and some tables lack necessary annotations. For instance, table 4, what do TE, PTE and SE stand for?

[Response 3]: Thank you for your patience in reviewing this paper. In response to the comments, We have provided a more detailed explanation of the icon title and description.

[Comments 4]�The use of terminology needs to be more consistent (such as mixing 'herding households' and 'herders').

[Response 4]: Thank you for your valuable comments. We changed "herding households" in the original text to "herders".

[Comments 5]�The policy recommendation section can be more specific and more closely related to research findings.

[Response 5]�We sincerely thank you for your insightful review and valuable comments on our findings. We have made more specific modifications to the policy recommendations section to make it more closely related to the research findings.

[Comments 6]�English writing requires further polishing, some sentences have grammatical errors and issues with expression fluency.

[Response 6]�Thank you for your insightful comments on the policy recommendations section. We have made more professional polishing of the language of the article to reduce grammatical errors existing in the paper and improve its fluency of expression. The specific modification contents can be found in the revision traces in the revised draft.

---

## [Editor Report · Decision Letter 1]

30 Sep 2025

Impact of the grassland ecological compensation policy on pastoral production efficiency—evidence from pastoral China

PONE-D-25-40838R1

Dear Dr. Zhang,

We’re pleased to inform you that your manuscript has been judged scientifically suitable for publication and will be formally accepted for publication once it meets all outstanding technical requirements.

Kind regards,

Abebayehu Aticho

Academic Editor

PLOS ONE
---

## [Editor Report · Acceptance letter]

PONE-D-25-40838R1

PLOS ONE

Dear Dr. zhang,

I'm pleased to inform you that your manuscript has been deemed suitable for publication in PLOS ONE. Congratulations! Your manuscript is now being handed over to our production team.

Kind regards,

on behalf of

Professor Abebayehu Aticho

Academic Editor

PLOS ONE